# SHREC: A framework for advancing next-generation computational phenotyping with large language models

Sarah Pungitore[1¤*], Shashank Yadav[2], Molly Douglas[1], Jarrod Mosier[1], Vignesh Subbian[2]

1 College of Medicine - Tucson, Tucson, Arizona, United States of America, 2 College of Engineering, The University of Arizona, Tucson, Arizona, United States of America

¤ Current address: College of Medicine - Tucson, The University of Arizona, Tucson, Arizona, United States of America

* spungitore@arizona.edu

## Abstract

Computational phenotyping is a central informatics activity with resulting cohorts supporting a wide variety of applications. However, it is time-intensive because of manual data review and limited automation. Since LLMs have demonstrated promising capabilities for text classification, comprehension, and generation, we posit they will perform well at repetitive manual review tasks traditionally performed by human experts. To support next-generation computational phenotyping, we developed SHREC, a framework for integrating LLMs into end-to-end phenotyping pipelines. We applied and tested three lightweight LLMs (Gemma2 27 billion, Mistral Small 24 billion, and Phi-4 14 billion) to classify concepts and phenotype patients using phenotypes for ARF respiratory support therapies. All models performed well on concept classification, with the best (Mistral) achieving an AUROC of 0.896. For phenotyping, models demonstrated near-perfect specificity for all phenotypes with the top-performing model (Mistral) achieving an average AUROC of 0.853 for single-therapy phenotypes. In conclusion, lightweight LLMs can assist researchers with resource-intensive phenotyping tasks. Several advantages of LLMs included their ability to adapt to new tasks with prompt engineering alone and their ability to incorporate raw EHR data. Future steps include determining optimal strategies for integrating biomedical data and understanding reasoning errors.

## Author summary

In our research, we explored how large language models like ChatGPT could help make the process of identifying patient groups from electronic health records faster and less labor-intensive. Traditionally, defining these patient groups requires

**Data availability statement:** All data used in this study are available through the eICU Collaborative Research Database (https://eicu-crd.mit.edu/). Access to this database is only granted to credentialed users. Interested researchers must complete registration and training steps as outlined on the eICU PhysioNet webpage (https://physionet.org/content/eicu-crd/2.0/).

**Funding:** SP supported in part by the National Institute of General Medical Sciences of the National Institutes of Health under grant T32 GM132008. https://cmmbs.arizona.edu/. The funders had no role in study design, data collection and analysis, decision to publish, or preparation of the manuscript.

**Competing interests:** The authors have no competing interests to declare.

careful manual review of large amounts of clinical data, which can be time-consuming and costly. We developed a framework called SHREC that integrates language models into these workflows, allowing the models to classify relevant clinical information and help create patient groups automatically. We tested several models on respiratory support therapies and found that even relatively small models were highly effective at accurately identifying concepts and patients. Our work shows that language models can complement human expertise, reducing the effort needed for routine tasks while still maintaining high accuracy. By demonstrating how these tools can fit into the larger research process, we hope to encourage further development of methods that make clinical data analysis faster, more efficient, and more accessible to researchers.

## Introduction

Computational, or electronic phenotyping, is a central informatics activity focused on defining, extracting, and validating meaningful clinical representations of digital data from electronic health records (EHRs) and other relevant information systems [1,2]. It is particularly fundamental to observational studies, large-scale pragmatic clinical trials, and healthcare quality improvement initiatives, where standardized, computable phenotypes allow for robust cohort discovery and monitoring of real-world outcomes [3]. Computable phenotypes have been developed for a wide variety of clinical outcomes and conditions, including acute conditions such as acute kidney injury [4], Acute Respiratory Distress Syndrome [5], and acute brain dysfunction in pediatric sepsis [6], and chronic conditions such as breast cancer [7], hypertension [8], and Post-Acute Sequelae of SARS-CoV-2 infection (PASC) [9]. They have also supported a variety of downstream tasks, including recruitment for clinical trials, development of clinical decision support systems, and hospital quality reporting [3,10,11].

The process of developing computable phenotypes typically includes identification and construction of relevant data elements for classification and then application of an algorithm to produce the cohort(s) of interest [12]. Traditionally, these processes involve multiple time and resource-intensive tasks requiring manual data review, such as mapping of data elements to controlled vocabularies [11]. Despite increased adoption of controlled vocabularies in EHR systems and improvements in Natural Language Processing (NLP) and machine learning methods, computational phenotyping remains complicated and costly [10,11]. As a result, many of the desiderata for phenotyping identified over a decade ago are still relevant today, indicating the need for substantial improvements to these methods [1,13]. To demonstrate these issues, we highlight challenges in development of computable phenotypes for PASC [9]. Since the phenotype definition was based on symptom presence, manual expert review of 6,569 concepts was first required to determine which were relevant to the 151 symptoms of interest. A series of data transformations were then applied to assess symptom presence relative to SARS-CoV-2 infection. Any new dataset, especially one not mapped to a controlled vocabulary, would require further manual review of concepts, rework of the algorithm, or both.

Given the existing opportunities with computational phenotyping and the minimal overall progress towards methodological improvements, it is natural to consider what will drive the next significant enhancement, or "next-generation" of phenotyping methods. In particular, with advances in machine and artificial intelligence, we also reconsider how much of the computational phenotyping process requires direct human involvement. The idea of human-machine synergy, with each component enhancing the abilities of the other, is fundamental to the field of informatics [14]. However, this synergy has yet to be achieved in computational phenotyping since humans still perform a majority of the phenotyping tasks, including ones where machines may excel. Therefore, we propose exploring the potential of Large Language Models (LLMs) for this domain. As a relatively new addition to biomedical research, LLMs introduce a novel set of text analysis, comprehension, and generation capabilities that allow them to analyze and generate text in ways that previously were either only possible by humans or by extensively trained, topic-specific NLP models [15]. Additionally, since LLMs are widely available as pre-trained foundation models, they can be adapted to new tasks through prompt engineering alone, which is a more accessible and portable method of model adaptation compared to model retraining or domain adaptation methods [16]. Furthermore, biomedical fine-tuned models underperformed on clinical tasks when compared to general-use LLMs, indicating that model retraining (a costly and time-intensive process) isn't even preferred for LLM adaptation [17]. Thus, the capabilities and advantages of LLMs satisfy many of the current deficits in computational phenotyping methods, suggesting their potential as foundational tools for next-generation phenotyping.

While some studies have applied LLMs to various clinical phenotyping tasks, none have explored the capability of LLMs to improve computational phenotyping specifically. The clinical phenotyping tasks studied include entity extraction and matching in clinical text [18,19], query generation for patient extraction [20], evaluation of hospital quality measures [21], and creation of phenotype definitions from standardized vocabulary codes [10]. When used to develop queries for identifying patients with type-2 diabetes mellitus, dementia, and hypo-thyroidism, GPT-4 produced queries that still required substantial oversight from human reviewers to generate accurate cohorts [20]. Additionally, when LLMs were used to generate computable phenotypes based on standardized vocabularies, GPT-4 only achieved an average accuracy of approximately 50% on both code matching and on string matching when compared to the original definition [10]. However, SOLAR 10.7B only slightly underperformed human categorizations for hospital quality measures and even provided a better response than human review in 4 out of 10 cases when responses between humans and LLMs differed [21]. Additionally, GPT-4o demonstrated perfect accuracy for classification of antibiotics from raw EHR data [22]. Therefore, while LLMs struggle with query and algorithm generation, even lightweight models have demonstrated the ability to categorize relevant clinical concepts from EHR data, further indicating potential application of LLMs for development of computable phenotypes.

Considering the opportunities in computational phenotyping methods and the novel capabilities of LLMs, we applied and evaluated LLMs to support computable phenotype development. We previously developed PHEONA (Evaluation of PHEnotyping for Observational Health Data), a framework specifically for evaluating LLMs for computational phenotyping tasks [23]. In this study, we expanded upon these methods to construct a broader view of next-generation phenotyping. The objectives of this study were thus the following:

1. Develop SHREC (SHifting to language model-based REal-world Computational phenotyping), a companion framework to PHEONA that outlines how to integrate LLMs into computational phenotyping.
2. Apply and demonstrate SHREC using previously developed computable phenotypes for Acute Respiratory Failure (ARF) respiratory support therapies.
3. Highlight future work and next steps to encourage progress towards next-generation phenotyping methods.

## Materials and methods

We first outline the development of SHREC along with its individual components (Fig 1 and Table 1) and then we discuss how we used LLMs to perform various tasks for a specific phenotyping use case.

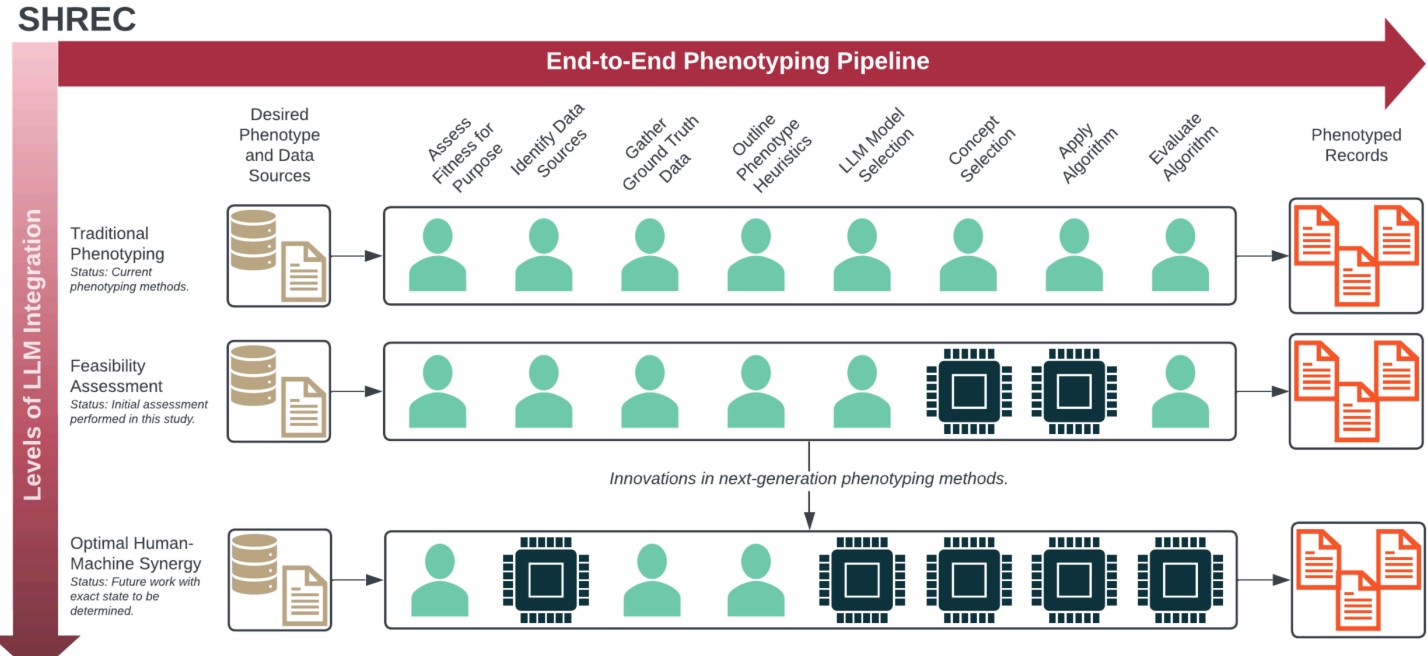

**Fig 1. Overview of SHREC.** An overview of SHREC (SHifting to language model-based REal-world Computational phenotyping), including both an overview of the individual phenotyping tasks and a representation of the progress required to advance next-generation computational phenotyping.

**Table 1. Overview of the individual end-to-end computational phenotyping tasks for SHREC (SHifting to language model-based REal-world Computational phenotyping), a framework for next-generation computational phenotyping with Large Language Model-based methods.** Some tasks were based on a previously developed framework for phenotype development using machine learning algorithms [12].

| Step | Name | Description | Analog to Previous Framework |
|---|---|---|---|
| 1 | Assess Fitness-for-Purpose | Determine the clinical outcome of interest, assess clinical significance, and assess any sources of clinical or data complexity. | Assess Fitness-for-Purpose |
| 2 | Identify Data Sources | Identify data source(s) to use for phenotype development and evaluation. | Assess Fitness-for-Purpose |
| 3 | Gather Ground Truth Data | Determine ground truth labels to use for validation of the phenotyping algorithm. | Create Gold Standard Data |
| 4 | Outline Phenotyping Heuristics | Determine the tasks necessary in the phenotyping process to obtain the resulting phenotypes from the input data. Will likely include the inclusion and exclusion criteria. | None |
| 5 | LLM Model Selection | If used, determine which LLMs can be tested for specific tasks and how these models can be evaluated [23]. For studies not using LLMs, can either skip or identify other machine learning or Natural Language Processing (NLP) models. | Develop Models |
| 6 | Concept Selection | Classify or identify relevant data elements from the electronic health record (EHR) data. | Engineer Features |
| 7 | Apply Algorithm | Apply the algorithm tasks to each record and identify the appropriate phenotype. | None |
| 8 | Evaluate Algorithm | Determine the effectiveness of the phenotyping algorithm against the ground truth data. | Evaluate Models |

## Development of SHREC

**Theoretical foundation.** To understand issues with computational phenotyping, we revisited the *Fundamental Theorem of Informatics*, which states optimized human-machine interactions should drive informatics methods [14], and *distributed cognition*, which describes how overall cognitive load is shared between internal and external agents [24,25]. In traditional phenotyping, humans generally could not delegate tasks to external agents without costly, time-consuming, or even impossible modifications [10,11,15]. Therefore, humans were responsible for both repetitive and complex tasks despite not being as inherently well-suited for repetitive work as machines. If LLMs are indeed capable of performing repetitive tasks well, these tasks can be offloaded to external LLM-based agents while ensuring humans are only responsible for complex ones to reduce overall cognitive burden of researchers and improve efficiency of the phenotyping process.

**Framework components.** Using this foundation, we constructed SHREC to include both an end-to-end phenotyping pipeline and a broader vision for next-generation computational phenotyping. To develop our end-to-end pipeline, we extended an existing framework originally developed for machine learning–based cohort discovery to more broadly capture computational phenotyping tasks [12]. Specifically, we added tasks for developing (*Outline Phenotyping Heuristics*) and implementing (*Apply Algorithm*) the phenotyping algorithm since they were previously implicit in development of the machine learning model. The end-to-end pipeline is detailed in Table 1. Meanwhile, the broader overview indicates overall progress towards optimal human-machine synergy in next-generation phenotyping (Fig 1). In this study, we only conducted a feasibility assessment of LLMs for specific phenotyping tasks.

## Application of SHREC to phenotyping use case

**Phenotyping use case.** We leveraged computable phenotypes for Acute Respiratory Failure (ARF) respiratory support therapies to demonstrate LLM-based methods for phenotyping tasks. Encounters were phenotyped based on the type and order of respiratory therapies received during individual Intensive Care Unit (ICU) encounters [26]. The phenotypes were 1) Invasive Mechanical Ventilation (IMV) only; 2) Noninvasive Positive Pressure Ventilation (NIPPV) only; 3) High-Flow Nasal Insufflation (HFNI) only; 4) NIPPV Failure (or NIPPV to IMV); 5) HFNI Failure (or HFNI to IMV); 6) IMV to NIPPV; and 7) IMV to HFNI.

**Identification of phenotyping tasks.** A comparison of the methods performed in the original study and this study are presented in Fig 2. In the original study, data from the eICU Collaborative Research Database (eICU-CRD) [27] were manually reviewed to first determine relevance to the therapies or medications of interest and second to produce a phenotyping algorithm [26]. These processes mapped to the *Concept Selection* and *Apply Algorithm* tasks of the end-to-end phenotyping pipeline within SHREC, respectively. In this study, we used LLMs to perform the tasks of *Concept Selection* and *Apply Algorithm* while we also implemented *Outline Phenotype Heuristics*, *LLM Model Selection*, and *Evaluate Algorithm* because they were required to execute and test the LLM-based methods. Since the remaining tasks were not related to the LLM methods, they were not included in this study, but were previously discussed in-depth [12,26].

## Implementation of LLMs for phenotyping tasks

The following sections detail the methods for each of the implemented phenotyping tasks from SHREC.

**Outline phenotype heuristics.** We determined the following heuristics from the previously developed algorithm [26]:

1. Identified the first encounter for each unique patient. Removed additional encounters to ensure a single encounter per patient.
2. Removed individuals less than 18 years old at the start of the encounter.

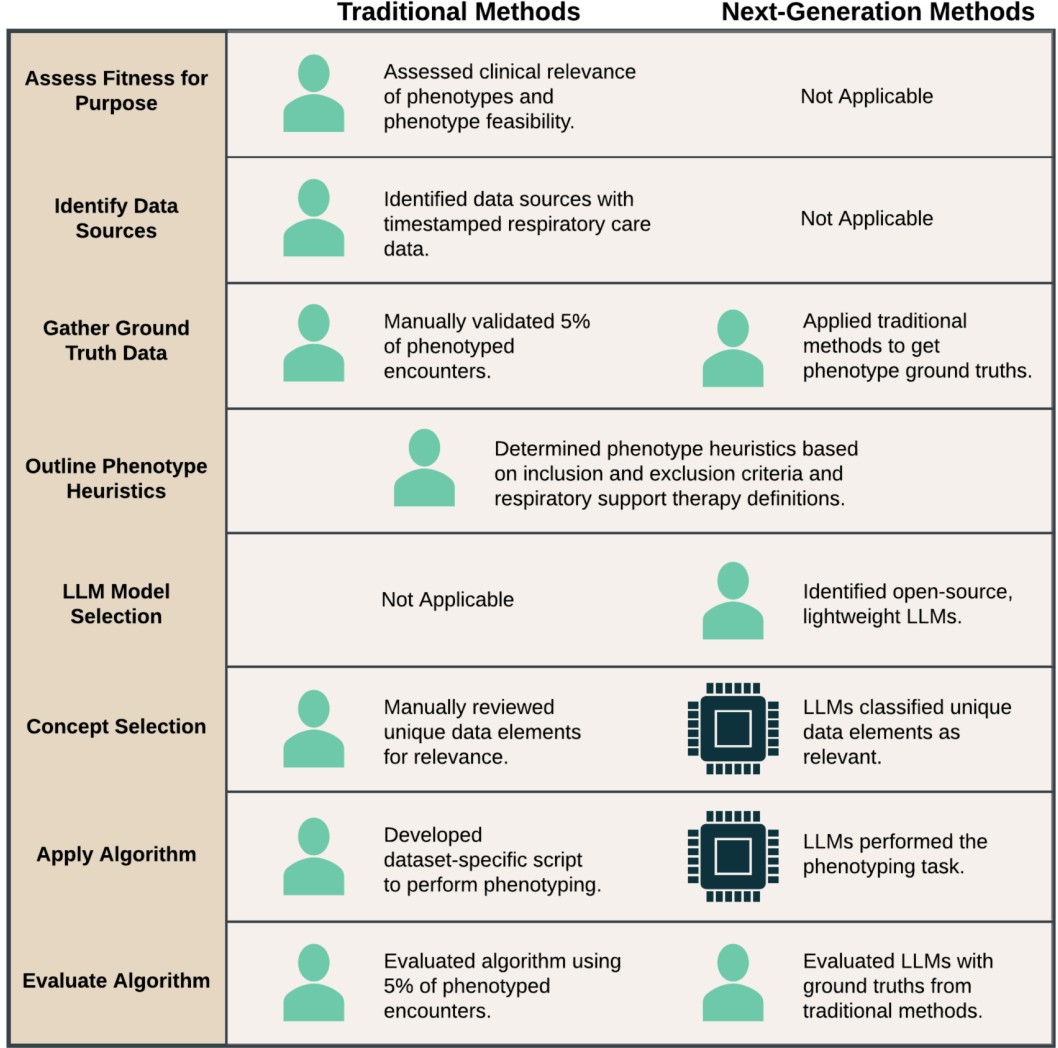

**Fig 2**. **Phenotyping methods comparison.** Comparison of traditional and next-generation methods for constructing phenotypes for Acute Respiratory Failure (ARF) respiratory support therapies, using the end-to-end phenotyping pipeline from SHREC (SHifting to language model-based REal-world Computational phenotyping). The traditional methods were used for initial phenotype development [26] while the next-generation methods were implemented in this study.

3. Extracted concepts from all distinct EHR records across all encounters and determined which were relevant to the respiratory support therapies (IMV, NIPPV, or HFNI) or medications (see Step 5*a*) of interest. For example, "*BiPAP/CPAP*" indicates NIPPV and "*Hi Flow NC*" indicates HFNI.

4. Filtered records for each encounter to only the extracted concepts for all of the respiratory support therapies and medications of interest.

5. Identified which of the respiratory support therapies were received during the encounter. The following criteria were used to determine if a therapy was received:

   (a) IMV: The presence of at least two records indicating IMV and at least one record indicating use of specific medication related to pre-intubation, intra-intubation, and post-intubation care (e.g., rapid sequence intubation medications, neuromuscular blocking agents, or continuous sedative agents).

(b)  NIPPV: At least two records indicating use of NIPPV AND no records indicating use of HFNI.

(c)  HFNI: The criteria for NIPPV is met AND there is at least one additional record indicating use of HFNI.

6. Determined the start and end for each treatment based on the offset time from ICU admission. When applicable, removed NIPPV and HFNI records that occurred between IMV records from consideration and reassessed whether criteria for NIPPV or HFNI was still met.

7. Classified any encounters where any of the respiratory support therapies were received into one of the following 8 phenotypes based on treatment criteria and ordering: 1) IMV only; 2) NIPPV only; 3) HFNI only; 4) NIPPV Failure (or NIPPV to IMV); 5) HFNI Failure (or HFNI to IMV); 6) IMV to NIPPV; 7) IMV to HFNI; and 8) No Therapies Received.

**LLM model selection.**  To promote reproducibility and adaptability of our methods, we selected LLMs available at the time of this study from Ollama, an open-source package that establishes local connections with open-source models [28]. Due to graphics processing unit (GPU) constraints, we selected the following lightweight, instruction-tuned models for testing: Mistral Small 24 billion with Q8.0 quantization (model tag: 20ffe5db0161), Phi-4 14 billion with Q8.0 quantization (model tag: 310d366232f4), and Gemma2 27 billion with Q8.0 quantization (model tag: dab5dca674db) [28]. DeepSeek-r1 32 billion with Q4_K_M quantization (model tag: 38056bbcbb2d) was previously tested for sampled data but was not used in this study due to high response latencies [23,28]. Models were run on a single Nvidia V100 32GB GPU. Temperature and top-p were 0.0 and 0.99, respectively, for all experiments to avoid responses outside the requested format.

**Concept selection.**  For *Concept Selection*, we generated constructed concepts from eICU-CRD tables. Tables that did not include timestamped data but contained information on respiratory therapies or medications (such as the *apacheApsVar* table) or were unlikely to contain descriptions of respiratory therapies or medications (such as the *vitalPeriodic* table) were not processed further. There were 9 tables used to construct input concepts (Table 2). In early testing, we achieved best results when classifying the respiratory therapies separately from the relevant medications. We thus developed two prompts, resulting in two LLM responses per constructed concept (see S1 Appendix). Concept definitions within the prompts were produced by two clinician experts who summarized and consolidated notes related to the terms of interest (see S1 Appendix). We used Chain-of-Thought (CoT) prompting by including a series of questions and answers to help the model determine relevancy of the constructed concept. CoT is a prompt engineering technique that has improved performance of LLMs by using a series of reasoning tasks to guide the model to the final response [29,30]. The answer to

**Table 2**. **Constructed concept pattern for each selected table in the eICU Collaborative Research Database database [27].**
Italicized text was replaced with values from the relevant column in each table, as demonstrated in the example for each table.

| Table Name | Constructed Concept Pattern | Example Constructed Concept |
|---|---|---|
| Care Plan General | Source = Care Plan General; Concept = *cplgroup*: *cplitemvalue* | Source = Care Plan General; Concept = Route-Status: Oral - low sodium |
| Infusion Drug | Source = Infusion Drug; Concept = *drugname* | Source = Infusion Drug; Concept = Amiodarone (mg/min) |
| Medication | Source = Medication; Concept = *drugname* | Source = Medication; Concept = LOPRESSOR |
| Note | Source = Note; Concept = *notevalue*: *notetext* | Source = Note; Concept = denies fevers: denies fevers |
| Nurse Care | Source = Nurse Care; Concept = *cellattributevalue* | Source = Nurse Care; Concept = emergency equipment at bedside |
| Nurse Charting | Source = Nurse Charting; Concept = *nursingchartcelltypevalname*: *nursingchartvalue* | Source = Nurse Charting; Concept = O2 Admin Device: BiPAP/CPAP |
| Respiratory Care | Source = Respiratory Care; Concept = *airwaytype* | Source = Respiratory Care; Concept = Oral ETT |
| Respiratory Charting | Source = Respiratory Charting; Concept = *respcharttypecat*: *respchartvaluelabel*: *respchartvalue* | Source = Respiratory Charting; Concept = respFlowPtVentData: SaO2: 25 |
| Treatment | Source = Treatment; Concept = *treatmentstring* | Source = Treatment; Concept = cardiovascular\|myocardial ischemia / infarction\|antiplatelet agent\|aspirin |

the final question for each prompt was parsed using string methods to get the final response of "YES" or "NO" for whether the constructed concept was relevant. Since there were two prompts, the final response was "YES" if at least one of the responses was "YES" and "NO" otherwise.

**Apply algorithm.** After we identified the relevant constructed concepts across the eICU-CRD dataset, we applied the phenotyping heuristics. We identified the first encounter for each unique patient and then removed individuals who were less than 18 years old at the start of the encounter. For each encounter, we filtered data from the original 9 tables to the selected constructed concepts and then ordered each distinct constructed concept by its first occurrence based on the encounter admission time. The data, or constructed descriptions, for phenotyping were created by inserting each individual constructed concept into a string template and concatenating all of the unique constructed concepts together for the prompt. The template was *"#: {constructed concept}"* where "#" was the order of the constructed concept based on its first occurrence in the encounter records. Since timestamps were generalized to concept order and there was no encounter-specific information in the constructed descriptions, we phenotyped the unique constructed descriptions and then mapped the phenotypes to the relevant encounters for evaluation. Our phenotyping prompt used CoT (see S1 Appendix). We parsed the answer to the final question to identify the selected phenotype.

**Evaluate algorithm.** Models were evaluated for both *Concept Selection* and *Apply Algorithm* using components of PHEONA, an evaluation framework for LLM-based applications to computational phenotyping [23]. Previously, we evaluated the models for *Concept Selection* using a random sample of constructed concepts [23]. In this study, we evaluated these models on the full set of constructed concepts using *Accuracy* (the ability of the model to produce accurate results) as the primary evaluation criterion and *Model Response Latency* (how quickly model results were returned) as the secondary evaluation criterion from PHEONA. We used area under the receiver operating characteristic curve (AUROC) to measure response accuracy against the concept ground truths. We measured response latency as the seconds required for the model to return a response for each constructed concept and then averaged these values for each prompt. For *Apply Algorithm*, since we had not previously used PHEONA for model evaluation, we evaluated model performance on a randomly selected subsample (see S1 Appendix) and then evaluated the best performing models on all encounters. We also used *Accuracy* and *Model Response Latency* criteria to assess model performance with the AUROC (and additionally, sensitivity and specificity) calculated using the original encounter ground truths [26].

## Results

### Phenotyping use case

There were initially 200,859 encounters across 166,355 patients. After applying the inclusion and exclusion criteria, there were 159,701 encounters for 159,701 patients. Using the previously developed phenotyping algorithm, the encounters were phenotyped as follows: 16,736 (10.5%) as IMV only; 6,833 (4.3%) as NIPPV only; 1,089 (0.7%) as HFNI only; 1,466 (0.9%) as NIPPV Failure; 568 (0.4%) as HFNI Failure; 601 (0.4%) as IMV to NIPPV; 186 (0.1%) as IMV to HFNI; and 132,222 (82.8%) as None [26].

### Implementation of LLMs for phenotyping tasks

**Concept selection.** There were 572 concept ground truths (404 ARF respiratory support therapies and 168 medications) from the original phenotyping study [26]. Classification results based on the concept ground truths are presented in Table 3. Mistral had the highest accuracy with an AUROC of 0.896 for classification of all concepts; however, it also had the highest total average latency (26.2 seconds compared to 22.0 seconds and faster). All models performed better at medication classification (AUROC of 0.997 and higher) when compared to respiratory support therapy classification (0.765 and higher).

**Apply algorithm.** There were 97,583 unique constructed descriptions for Gemma, 62,499 for Mistral, and 65,581 for Phi. However, since Gemma underperformed on the subsample of constructed descriptions (see S1 Appendix), only

**Table 3**. **Results of *Concept Selection* using Gemma2 27 billion, Mistral Small 24 billion, and Phi-4 14 billion Large Language Model models.** The number of concepts selected, area under the receiver operating characteristic curve (AUROC), and average latency (measured in seconds) were measured for both the Acute Respiratory Failure (ARF) respiratory support therapies and medications prompts.

| Model | Total Concepts | | | ARF Support Therapies Concepts | | | Medications Concepts | | |
|---|---|---|---|---|---|---|---|---|---|
| | N | AUROC[a] | Latency | N | AUROC[a] | Latency | N | AUROC[a] | Latency |
| Gemma | 30,062 | 0.792 | 22.0 | 29,394 | 0.783 | 17.7 | 674 | 0.997 | 4.4 |
| Mistral | 7,143 | 0.896 | 26.2 | 6,754 | 0.872 | 16.6 | 389 | 0.996 | 9.6 |
| Phi | 13,829 | 0.809 | 19.0 | 13,397 | 0.765 | 12.2 | 433 | 0.998 | 6.8 |

[a] AUROC: Area under the receiver operating characteristic curve.

Mistral and Phi were tested on the entire dataset. Mistral had an average response latency of 27.3 seconds and Phi of 20.8 seconds across all constructed descriptions. The AUROC, sensitivity, and specificity for each phenotype are presented in Table 4. Overall, Mistral performed better with a higher AUROC on all phenotypes when compared to Phi. Mistral performed best on no therapy and single therapy phenotypes (None and IMV, NIPPV, and HFNI Only) with an average AUROC of 0.853 while it only achieved an average AUROC of 0.604 on the remaining, multi-therapy phenotypes. Both models also had nearly perfect specificity for all phenotypes except IMV Only.

## Discussion

In this study, we introduced SHREC, a framework for applying LLM-based methods to computational phenotyping. We outlined the components of SHREC and demonstrated how LLMs can be used for computational phenotyping tasks.

### Development and application of SHREC

The primary contribution of this study was the components of SHREC. For the first component, the end-to-end phenotyping pipeline, we expanded upon a previously developed framework and generalized it to apply to all computable phenotypes, not just those involving either machine learning or LLMs [12]. For the second component, we outlined a novel system of LLM-based next-generation computational phenotyping. In its future state, we envision all repetitive tasks

**Table 4**. **Results of *Apply Algorithm* using Mistral Small 24 billion and Phi-4 14 billion Large Language Models.** The number of phenotyped encounters, area under the receiver operating characteristic curve (AUROC), sensitivity, and specificity were measured for both models across 159,701 encounters.

| Phenotype | N | Mistral | | | | Phi | | | |
|---|---|---|---|---|---|---|---|---|---|
| | | N | AUROC[a] | Sens.[b] | Spec.[c] | N | AUROC[a] | Sens.[b] | Spec.[c] |
| IMV[d] Only | 16,736 | 33,537 | 0.881 | 0.898 | 0.864 | 36,343 | 0.863 | 0.936 | 0.791 |
| NIPPV[e] Only | 6,833 | 7,060 | 0.809 | 0.637 | 0.981 | 7,158 | 0.758 | 0.547 | 0.969 |
| HFNI[f] Only | 1,089 | 2,325 | 0.825 | 0.661 | 0.989 | 843 | 0.543 | 0.093 | 0.994 |
| NIPPV[e] Failure | 1,466 | 2,063 | 0.717 | 0.443 | 0.991 | 1,463 | 0.669 | 0.347 | 0.992 |
| HFNI[f] Failure | 568 | 377 | 0.513 | 0.028 | 0.998 | 208 | 0.503 | 0.007 | 0.998 |
| IMV[d] to NIPPV[e] | 601 | 1,725 | 0.526 | 0.063 | 0.989 | 1,536 | 0.565 | 0.143 | 0.987 |
| IMV[d] to HFNI[f] | 186 | 1,545 | 0.659 | 0.328 | 0.990 | 1,026 | 0.539 | 0.086 | 0.991 |
| None | 132,222 | 103,990 | 0.896 | 0.824 | 0.968 | 66,952 | 0.845 | 0.744 | 0.947 |

[a] AUROC: Area under the receiver operating characteristic curve.
[b] Sens.: Sensitivity.
[c] Spec.: Specificity.
[d] IMV: Invasive Mechanical Ventilation.
[e] NIPPV: Noninvasive Positive Pressure Ventilation.
[f] HFNI: High-Flow Nasal Insufflation.

(including manual data review) being offloaded to LLM agents [31] while humans guide complex tasks and provide deliberate oversight to best satisfy the *Fundamental Theorem of Informatics* [14]. Towards this end, we noted several key properties of LLM-based methods that would support widespread integration into the end-to-end pipeline. First, prompt engineering alone was sufficient for adapting the models to both *Concept Selection* and *Apply Algorithm* without additional retraining or algorithm development. Second, minimal data processing was required: other than tagging concepts with the original table name and recording order, raw EHR data were used for both tasks. Therefore, even the lightweight models tested in this study demonstrated clear advantages over traditional phenotyping methods, including advanced NLP and machine learning algorithms.

## Implementation of LLMs for phenotyping tasks

The second contribution of this study was the demonstration of LLMs for the tasks of *Concept Selection* and *Apply Algorithm* from the end-to-end pipeline. All models performed well at concept classification, especially classification of the medications concepts (Table 3). For the phenotyping tasks, Mistral and Phi generally performed better at determining phenotypes with only a single treatment when compared to those with a sequence of treatments (Table 4). We suspect the layered thought process of assigning records to a treatment and then determining treatment order was too complex for the models tested. We hypothesize that either mapping each constructed concept to a specific treatment or performing a second phenotyping step solely for treatment ordering would improve phenotyping performance. These results suggest that lightweight LLMs can be readily applied to concept classification and simple phenotypes but may currently be insufficient for complex phenotypes without enhancements to the base models, prompts, or pipeline within *Apply Algorithm*.

One outstanding question for all biomedical tasks performed with LLMs is how to best incorporate specialized medical knowledge, including standardized vocabularies and ontologies. In this study, we injected medical information into the prompts and relied on the inherent capabilities of each model for data synthesis and comprehension. This method is applicable to many other computable phenotypes, including the previously discussed phenotypes for PASC where the Observational Medical Outcomes Partnership Common Data Model (OMOP-CDM) concepts could be categorized by injecting prompts with symptom information [9]. Outside of computational phenotyping, studies have also explored methods for providing SNOMED CT knowledge to LLMs, including prompt injection, model pretraining, and model finetuning, although almost none of these studies reported performance results after incorporating SNOMED CT [32]. Furthermore, a recent study found finetuned biomedical LLMs underperformed when compared to generalist models on multiple clinical benchmarking tasks [17]. Therefore, while there is some evidence that prompt injection (with or without retrieval augmented generation) may be the best method for incorporating domain knowledge, our results suggest there may be limits to the effectiveness of this method. Thus, there remain significant gaps in understanding how to best incorporate specialized medical knowledge into LLMs both for general biomedical and computational phenotyping tasks.

## Study limitations

There were several limitations to the framework and methods implemented in this study. First, repetitive manual review was still required for model evaluation. Furthermore, since we used previously developed ground truths, we performed less manual review than would be required for a novel phenotyping study. However, manual review for LLM-based methods can be reduced by reviewing samples rather than the entire dataset. Another limitation is scalability. For *Concept Selection* and *Apply Algorithm*, each constructed concept and description required individual LLM responses and thus, after a certain number of records, it would become infeasible to use LLMs. However, for this study, almost a sixfold increase in records would be required before the time for LLM-based phenotyping became comparable to the time for traditional phenotyping methods. Additionally, as improvements in model performance, retrieval augmented generation architecture, and prompt engineering are developed, we expect the issue of scalability for LLM-based methods to be lessened, although not completely mitigated.

### Future directions

There are many future directions to explore outside of those highlighted by the broader vision of computational phenotyping from SHREC. One direction is to understand how LLMs reason with respect to computational phenotyping. Although we used CoT because of its demonstrated ability to produce accurate results [29,30,33], recent studies have suggested that CoT reasoning may actually be unfaithful [34–36]. Given the complexity of phenotyping tasks, we suggest studying CoT reasoning to understand when and how logical inconsistencies may arise. Another future direction is to adapt phenotype definitions to LLMs. For example, the clinician experts indicated some of the medications may be present for non-invasive therapies, but not as a continuous infusion (see S1 Appendix). However, for the previously developed definition, only medication presence was used due to complexities in algorithmically processing medication information in relation to respiratory therapies. In future iterations, we would update the phenotype definition to include administration method rather than simply asking the model to look for concept presence. Finally, we propose development of industry standards for evaluation of LLM-based methods specifically with respect to automated processes to ensure appropriate oversight.

### Conclusion

We developed SHREC, a framework that describes how to apply LLM-based methods to computational phenotyping. SHREC outlines both an end-to-end pipeline for computable phenotype development along with a broader vision of next-generation phenotyping using LLM-based methods. We demonstrated SHREC on a phenotyping use case to assess the feasibility of LLMs for specific phenotyping tasks and promote further research into next-generation computational phenotyping methods. This work is applicable to all computational phenotyping studies, particularly those using manual review for phenotype development.

### Supporting information

**S1 Appendix. Supplementary material.** The supplementary material file contains prompts, concept notes, the application of PHEONA (Evaluation of PHEnotyping for Observational Health Data) to the Apply Algorithm step, and details on code availability.
(PDF)

### Acknowledgments

This work was supported in part by the National Institute of General Medical Sciences of the National Institutes of Health under grant T32 GM132008.

### Author contributions

**Conceptualization:** Sarah Pungitore, Shashank Yadav, Molly Douglas, Jarrod Mosier, Vignesh Subbian.

**Data curation:** Sarah Pungitore.

**Formal analysis:** Sarah Pungitore.

**Investigation:** Sarah Pungitore.

**Methodology:** Sarah Pungitore, Shashank Yadav.

**Resources:** Sarah Pungitore.

**Software:** Sarah Pungitore.

**Supervision:** Vignesh Subbian.

**Validation:** Sarah Pungitore.

**Visualization:** Sarah Pungitore.

**Writing – original draft:** Sarah Pungitore.

**Writing – review & editing:** Sarah Pungitore, Shashank Yadav, Molly Douglas, Jarrod Mosier, Vignesh Subbian.

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
