## [Decision Letter · Decision Letter 0]

20 Jan 2026

SHREC: A framework for advancing next-generation computational phenotyping with large language models

PDIG-D-25-00611

Dear Dr. Pungitore,

We are pleased to inform you that your manuscript 'SHREC: A framework for advancing next-generation computational phenotyping with large language models' has been provisionally accepted for publication in PLOS Digital Health.

Best regards,

Marie-Laure Charpignon, MSc

Academic Editor

PLOS Digital Health

**Additional Editor Comments (if provided):**

Dear Authors,

Thank you for submitting your high-quality research to PLOS Digital Health.

I recommend your manuscript for publication.

Two of the reviewers shared several recommendations for improvement. One reviewer also posted a reflection question for you to consider.

Provided you address the few weaknesses listed by the reviewers and answer their specific questions, I think your review article would constitute a great fit for the journal.

Thank you in advance for your edits and contributions,

Marie.

**Reviewer Comments (if any, and for reference):**

Reviewer's Responses to Questions

**Comments to the Author**

1. Does this manuscript meet PLOS Digital Health’s publication criteria? Is the manuscript technically sound, and do the data support the conclusions? The manuscript must describe methodologically and ethically rigorous research with conclusions that are appropriately drawn based on the data presented.

Reviewer #1: Yes

Reviewer #2: Yes

Reviewer #3: Yes

2. Has the statistical analysis been performed appropriately and rigorously?

Reviewer #1: Yes

Reviewer #2: Yes

Reviewer #3: Yes

3. Have the authors made all data underlying the findings in their manuscript fully available (please refer to the Data Availability Statement at the start of the manuscript PDF file)?

Reviewer #1: Yes

Reviewer #2: Yes

Reviewer #3: Yes

4. Is the manuscript presented in an intelligible fashion and written in standard English?

Reviewer #1: Yes

Reviewer #2: Yes

Reviewer #3: Yes

5. Review Comments to the Author

Reviewer #1: Thank you for preparing an interesting paper outlining your new approach computational phenotyping using LLM's. You have carried out a comprehensive review along with a novel framework for improving on the work of computational phenotyping.

Reviewer #2: In this paper, the authors introduce the SHREC framework for computational phenotyping with LLMs, and include evaluations of 3 LLMs on several components of the pipeline.

Strengths:

The paper is well-written and easy to follow, and the framework is intuitive. The authors also show how some manual curation + LLM-based extraction can help this use case. The high AUROCs for each of the tasks is also interesting. The code and study setup has also been made available.

Weaknesses:

It is unclear if the authors have reported results with multiple runs, and have kept the prompt etc fixed across the models. Also it would be interesting to test a few models specialized in the medical domain for this task (e.g., domain specific LLMs).

Misc.:

The authors alluded to this in the discussion, but performing some tradeoff analysis in terms of latency, both computationally and with a study with doctors/clinicians in the loop could further strengthen this work.

Questions:

Can authors comment on how "difficult" some of the tasks are? For example, there is a wide variation in the concept extraction for the two sub-types. Does this align with human dificulty as well? If there are specific tasks within SHREC that are more time consuming/difficult via manual curation but well-performing with LLMs, that would be another advantage of such a framework.

Reviewer #3: Summary of the Work

This paper introduces SHREC, a framework for using Large Language Models in computational phenotyping. The authors demonstrate their approach by applying three lightweight LLMs to classify acute respiratory failure phenotypes from electronic health record data. The main contribution is showing that LLMs can automate manual review tasks that traditionally require substantial human effort. The paper is well-written and addresses an important problem in clinical informatics where manual concept review creates significant bottlenecks in phenotype development.

Main Claims and Significance

The paper claims that LLMs can effectively replace manual review tasks in computational phenotyping, particularly for concept selection and algorithm application. This is significant because manual review is time-consuming and costly. The authors tested three models on respiratory therapy phenotypes and found that Mistral achieved the highest accuracy with an AUROC of 0.896 for concept classification. For phenotyping tasks, models performed better on single therapy phenotypes compared to multi-therapy phenotypes.

The significance for the field is high because if LLMs can reliably automate these tasks, it would substantially reduce the time and resources needed for observational studies and clinical trials. However, the paper would be stronger if it directly compared LLM performance to the original manual review performance. Currently the paper evaluates LLMs against ground truth labels but does not tell us how well the human experts performed on these same tasks. Including inter-rater reliability metrics from the original manual review and comparing LLM accuracy to human expert accuracy would help readers understand if LLMs truly achieve comparable performance.

Literature Context and Novelty

The introduction does a good job motivating the problem of manual review in phenotyping. The authors cite relevant work on computable phenotypes for various conditions and discuss previous applications of LLMs to clinical tasks. However, the distinction between this work and prior LLM phenotyping studies could be clearer. Several cited papers appear to address similar computational phenotyping tasks, making it difficult to understand what is genuinely novel about SHREC versus existing approaches.

It would help to add a clear statement explaining what makes SHREC different from prior work. For example, is it the end-to-end pipeline approach, the focus on lightweight models, the specific combination of tasks, or something else? The relationship between PHEONA and SHREC also needs clarification. From reading the paper it seems PHEONA is for evaluation while SHREC is for implementation but this should be stated explicitly early on.

The discussion of domain knowledge integration is thoughtful and the authors appropriately acknowledge that incorporating medical vocabularies and ontologies into LLMs remains an open question. The references to distributed cognition and the Fundamental Theorem of Informatics provide good theoretical grounding for the work.

Data and Statistical Analysis

The results section presents clear performance metrics and appropriately reports the class distribution showing that most encounters had no respiratory therapy. This transparency about class imbalance is appreciated. However, several aspects of the statistical analysis need strengthening.

First, the paper reports AUROC values but does not include confidence intervals. Adding 95 percent confidence intervals for all reported AUROC values would help readers assess the precision of these estimates. Second, there is no statistical testing to determine if performance differences between the three models are significant. Including statistical tests such as the DeLong test for comparing AUROCs would strengthen the conclusions.

Third, the paper would benefit from more detailed error analysis. The observation that multi-therapy phenotypes achieved much lower AUROC compared to single-therapy phenotypes is interesting but the paper does not provide examples of what types of errors occurred. Adding a confusion matrix or specific examples of misclassifications would help readers understand the failure modes and could guide future improvements to the approach.

Methods and Reproducibility

The methods section provides excellent technical detail including model versions, quantization levels, hardware specifications, and hyperparameters. This level of detail supports reproducibility. However, several key pieces of information are missing or incomplete.

The paper does not describe how ground truth labels were established. Were the original concepts manually annotated by clinical experts? How many annotators were involved? What was the annotation process? What was the inter-rater agreement? This information is essential for understanding the quality of the evaluation standard.

The paper references prompts multiple times but indicates they are in the supplementary appendix. Given that prompt engineering is central to the LLM approach and is highlighted as a key advantage in the introduction, it would be helpful to include at least one example prompt in the main text. Additionally, the authors should explicitly commit to releasing the full prompts and code publicly to support reproducibility.

Generalizability and Scope

The paper tests SHREC on one clinical use case involving acute respiratory failure phenotypes with temporal treatment sequences. While this is a solid proof of concept, the generalizability to other phenotype types is unclear. The discussion would be strengthened by addressing which types of phenotypes SHREC is best suited for. For example, would it work for phenotypes requiring interpretation of lab values or vital signs?

The authors acknowledge that lightweight models struggled with complex multi-therapy phenotypes. This is an honest and important observation. Expanding on this to help readers understand the current limitations of LLM capabilities for phenotyping would be valuable. The authors suggest that either mapping concepts to treatments first or performing phenotyping in two steps might improve performance. These are good ideas and could be highlighted as specific next steps for future work.

Clinical Translation and Impact

The paper positions SHREC as next-generation phenotyping but does not discuss practical considerations for clinical translation. How would researchers or clinicians validate LLM-generated phenotypes before using them in studies? What level of oversight is needed? The paper mentions that manual review is still required for model evaluation, which is true for any validation study, but it would help to clarify whether after initial validation, LLMs could reduce manual review for new phenotypes or new datasets.

The paper claims LLMs demonstrate clear advantages over traditional phenotyping methods including advanced NLP and machine learning algorithms. However, the study did not directly compare LLMs to these traditional methods. This claim would be more convincing with either direct comparison data or softer language such as potential advantages or comparable performance with additional benefits like flexibility and portability.

Discussing computational costs would also be helpful. The authors mention that a sixfold increase in records would be required before LLM-based phenotyping became comparable in time to traditional methods, but actual time comparisons would make this more concrete. For example, if manual review took a certain number of hours and LLMs took a different number of hours, stating this explicitly would help readers assess practical feasibility.

Limitations

The limitations section is honest and discusses important points including the need for manual review during evaluation and scalability concerns. However, several additional limitations should be acknowledged.

First, testing was limited to a single dataset and single clinical domain. External validation on other datasets or phenotypes would strengthen confidence in generalizability.

Second, the paper does not discuss potential LLM hallucinations or the risk of models generating plausible but incorrect classifications. This is a known issue with LLMs and should be acknowledged as a consideration for clinical applications.

Third, interpretability and explainability are not discussed. When LLMs misclassify a phenotype, understanding why the error occurred is important for clinical users. Chain of thought prompting provides some reasoning transparency but this could be discussed more.

Results Reporting

The paper reports nearly perfect specificity for most phenotypes but sensitivity values should also be prominently reported. High specificity with lower sensitivity would suggest the models are overly conservative and might miss true cases. For clinical applications, understanding this trade-off is important.

Organization and Writing

The paper is well organized and clearly written making it accessible to readers who may not be computational phenotyping experts. The theoretical framework grounding in distributed cognition adds depth and helps readers understand the motivation beyond just technical capability. The figures comparing traditional versus next-generation methods are helpful for visualizing the approach.

Overall Assessment

This is valuable work addressing an important problem in clinical informatics. The SHREC framework is well conceived and the technical execution demonstrates that lightweight LLMs can perform phenotyping tasks with promising accuracy. The work has strong potential to advance computational phenotyping methods and make phenotype development more efficient and accessible.

The main areas for improvement are adding comparison to human performance, including statistical testing and confidence intervals, documenting ground truth development methods, providing example prompts, and strengthening discussion of generalizability and clinical translation. These revisions are straightforward and do not require new experiments, primarily involving additional analysis of existing data and clearer documentation.

6. PLOS authors have the option to publish the peer review history of their article (what does this mean?). If published, this will include your full peer review and any attached files.

**Do you want your identity to be public for this peer review?** For information about this choice, including consent withdrawal, please see our Privacy Policy.

Reviewer #1: No

Reviewer #2: No

Reviewer #3: No
